# A Model of Integrated Community-Based Bamboo Management for the Bamboo Industry in Ngada Regency, East Nusa Tenggara, Indonesia

**Desy Ekawati** [1,2,*] **, Lina Karlinasari** [1,3,*] **, Rinekso Soekmadi** [4] **and Machfud** [5]

1  Study Program of Natural Resource and Environmental Management Science, Graduate School, IPB University, Baranangsiang Campus, Bogor 16153, Indonesia
2  Standardization Instrument of Environment and Forestry Agency, Ministry of Environment and Forestry, Bogor 16118, Indonesia
3  Department of Forest Products, Faculty of Forestry and Environment, IPB University, Darmaga Campus, Bogor 16680, Indonesia
4  Department of Forest Resource Conservation and Ecotourism, Faculty of Forestry and Environment, IPB University, Darmaga Campus, Bogor 16680, Indonesia
5  Department of Agroindustry Technology, Faculty of Agricultural Technology, IPB University, Darmaga Campus, Bogor 16680, Indonesia
*  Correspondence: desy@apps.ipb.ac.id (D.E.); karlinasari@apps.ipb.ac.id (L.K.)

**Abstract:** The potentials of bamboo resources owned by the community in Ngada Regency has not been managed and appropriately utilized. There were no integrated programs between the on-farm and off-farm sectors and no clear roles and responsibilities among the stakeholders involved. Soft System Methodology (SSM) framework approach was carried out through stakeholder analysis, CATWOE analysis, and gap analysis. The root definition of the current situation was that the model of sustainable community bamboo management and utilization (W) is responsible to the local and central government as well as the bamboo manufacturing industry as off-taker (O) with integrated supporting programs and regulations, ensuring the potential of bamboo resources and the bamboo product market (E) which was carried out together with stakeholders (A) through active participation and synergy programs (T) to improve the welfare of the community of bamboo owners, craftsmen, and bamboo entrepreneurs (C). The study produces a suitable and appropriate strategy based on the corrective actions of existing problems and recommendations formulated from conceptual models and existing actual conditions on integrated sustainable bamboo management.

**Keywords:** bamboo resources; SSM framework; CATWOE analysis; stakeholder analysis; sustainable management; integrated strategy

## 1. Introduction

Bamboo is a natural resource with great potential which is yet to be optimally utilized. As a natural resource close to the community, especially for those living in rural areas, bamboo can be a commodity driving the economy of households and rural communities while providing ecological and environmental functions [1,2]. The increasing population, which underpins the rising need for housing and agricultural land, increases suppressing the bamboo cover in rural areas. Bamboo resources are yet to be a priority in their management and utilization. The use of bamboo, in general, and its management challenges in Indonesia still need to be connected between the parties and actors involved. Optimal management and utilization of bamboo require synergy and integration between on-farm and off-farm activities. The connection between surveillance, production, and processing activities supported by strong markets and value chains will drive the sustainable use of community-based bamboo resources [3]. Community-based bamboo development can be a force to boost rural development by advancing the bamboo industry with value-added

products, a lesson learned from the success of bamboo development in China [4,5]. In China, smallholder farmers enthusiastically embraced bamboo as a new cash crop and planted it and other non-timber forest crops on their designated forest land instead of less-profitable timber species [6–8]. In line with developments upstream, the bamboo processing industry began to grow, so today, the bamboo industry has become a pillar of the rural economy generation of China [9]. In recent years, the development of technology and innovation has changed the use and utilization of bamboo. Bamboo has become a noble material that can be processed into multiple products with various uses. Advanced technology in construction will contribute to the development and expansion of bamboo applications (buildings, structures, bridges, etc.) [10].

In Ngada Regency, East Nusa Tenggara Province, Indonesia, the use of community bamboo as a supplier of raw materials regularly to the bamboo strips processing industry has been running. In 2012, a bamboo processing factory was established, which produces preserved bamboo strips and sticks, which will be further processed into engineered bamboo products [11]. This factory processes community bamboo by requiring bamboo clumps management and selected harvests of four-year-old bamboo stems with sustainable bamboo forestry (SBF) [12]. Then, in 2016, the Ministry of Environment and Forestry (MoEF), the Republic of Indonesia, in collaboration with the Environmental Bamboo Foundation (EBF), initiated the bamboo village movement, a community-based integrated bamboo industry development platform. Through this movement, Ngada Regency has become a pilot project and a pilot for community-based integrated bamboo development from upstream to downstream through the involvement of various relevant stakeholders [11]. However, it is still facing multiple obstacles from technical and non-technical aspects and policy and regulatory support.

Therefore, this study was conducted to develop a model of integrated community-based bamboo management for the bamboo industry in Ngada Regency based on the actual condition and problematic situation. The primary purposes of this study were to construct an adaptive and appropriate management model and to find suitable strategies for community-based sustainable bamboo management and utilization. To realize this, the research objectives that need to be elaborated were as follows: (i) identifying related stakeholders and institutions as well as their roles and responsibility, (ii) obtaining key elements that will strengthen community-based bamboo management, and (iii) formulating appropriate strategies, as corrective actions and recommendation to create community-based bamboo management and its utilization.

## 2. Methods

### 2.1. Study Area

Ngada Regency is located in the central part of Flores Island of East Nusa Tenggara Province, Indonesia (8°20′24.28″–8°57′28.39″ south latitude and 120°48′29.26″–121°11′8.57″ east longitude (Figure 1)). The area is 1776.72 km$^2$, with a population of 165,254 inhabitants and a growth rate of 1.25% per year. Stretching from the south, there are 12 districts in Ngada regency, and the capital is Bajawa city, which is the most populated area in the southern part [13]. Besides being famous for bajawa arabica coffee and candlenuts, Ngada has a high potential for bamboo resources. Bamboo has also become part of their culture and livelihood over generations. This is shown by their traditional and tribe's house and also local wisdom. Ngada is also famous for its megalith culture [14] and was added to the UNESCO World Heritage Tentative List on 19 October 1995, in the cultural category.

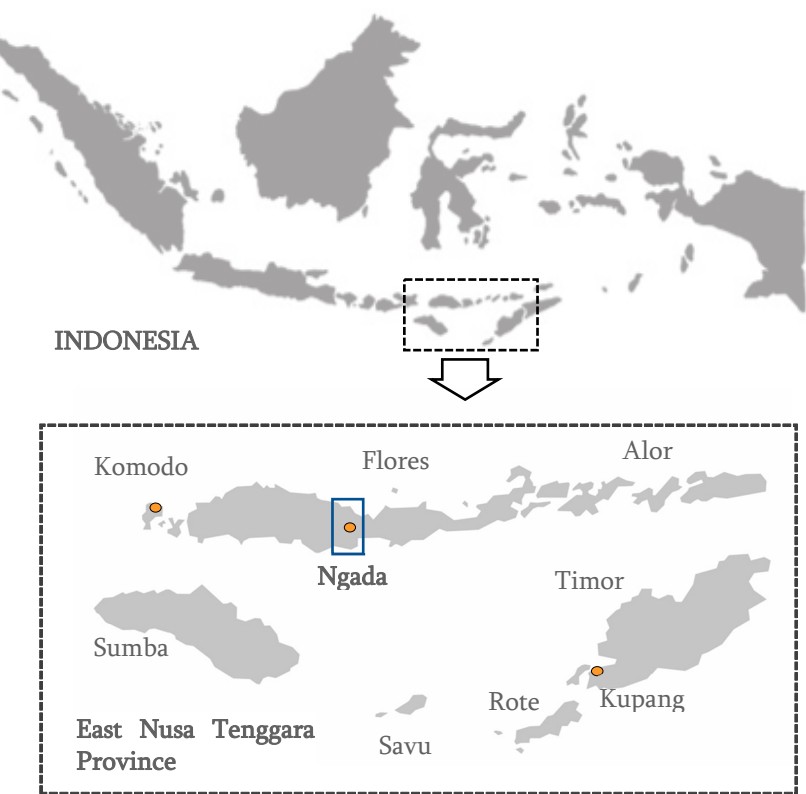

**Figure 1.** Study site and location.

Ngada's income mainly comes from the agricultural sector, especially fisheries and forestry. This sector accounts for 34.15% of the total Gross Regional Domestic Product (GDRP), followed by the public sector (19.88%) and the construction sector (13.27%) [15]. Other contributing economic sectors are wholesale and retail traders and car and bike repair shops.

Regarding landscape management, this site represents a highland with mountainous and hilly regions. It receives high rainfall at 1500–2000 mm/year intervals due to the surrounding oceans and the differences in air pressure and temperatures. Generally, the community uses water sources from the springs of the surrounding mountainous region from their villages [16]. However, lack of clean water occurs in areas far from springs and some areas where the ecosystem and habitat are degraded and disturbed. The Ngada area is also upstream of the Aesesa watershed, including the Wulabhara and Wae Woki watersheds, with rivers streaming down the north and the south coast [17].

*2.2. Research Framework and Stages*

The Soft System Methodology (SSM) framework was used to build a model for the community's sustainable bamboo management. This framework was chosen to understand the problems of the situation faced by the stakeholders concerned by designing a system of human activity to improve the role of participation. By understanding and having common issues, the stakeholders could commit to making changes and achieving the goals [18,19].

Therefore, to develop an adaptive integrated management model as well as to find suitable strategies for community-based sustainable bamboo utilization, this study identified the seven stages of the SSM framework (Figure 2), as follows:

1. Identifying the current conditions and problems related to stakeholders involved in community bamboo management and utilization. In this early stage, there is a focus on the accurate site actual that exists in the use of community bamboo in Ngada Regency and the role and interconnection of stakeholders involved in it, from policy aspects issued by the government to technical levels in business actors.

2. Understanding the problems and situations stakeholders face given their needs, roles, and responsibilities. This stage enables the creation of rich pictures that depict interconnections between the issues faced by stakeholders and the current situation.

3. Defining the role and influences of each group and stakeholder based on an approach called client or customer, actor, transformation, world view, owner, and environmental constraints (CATWOE).

4. Designing a conceptual model that describes the activities and interconnection between the actions needed to synthesize the best solution for sustainable community-based bamboo management. Additionally, this step enabled us to find key elements that significantly influence the strengthening efforts based on in-depth interviews with key actors.

5. Composing the arrangement of actual activities and then comparing those to the conceptual model using gap analysis.

6. Defining possible changes, including procedures, structures, and cultures in the form of values, norms, and ways of thinking by confirming to the key actors involved. Such changes also occurred by developing strategic assumptions of attempts to build the integrated management model of community bamboo utilization for the industry.

7. Formulating an appropriate strategy for implementing integrated management and corrective actions based on assumptions and recommendations based on possible changes from the developed model's comparison and the actual situation.

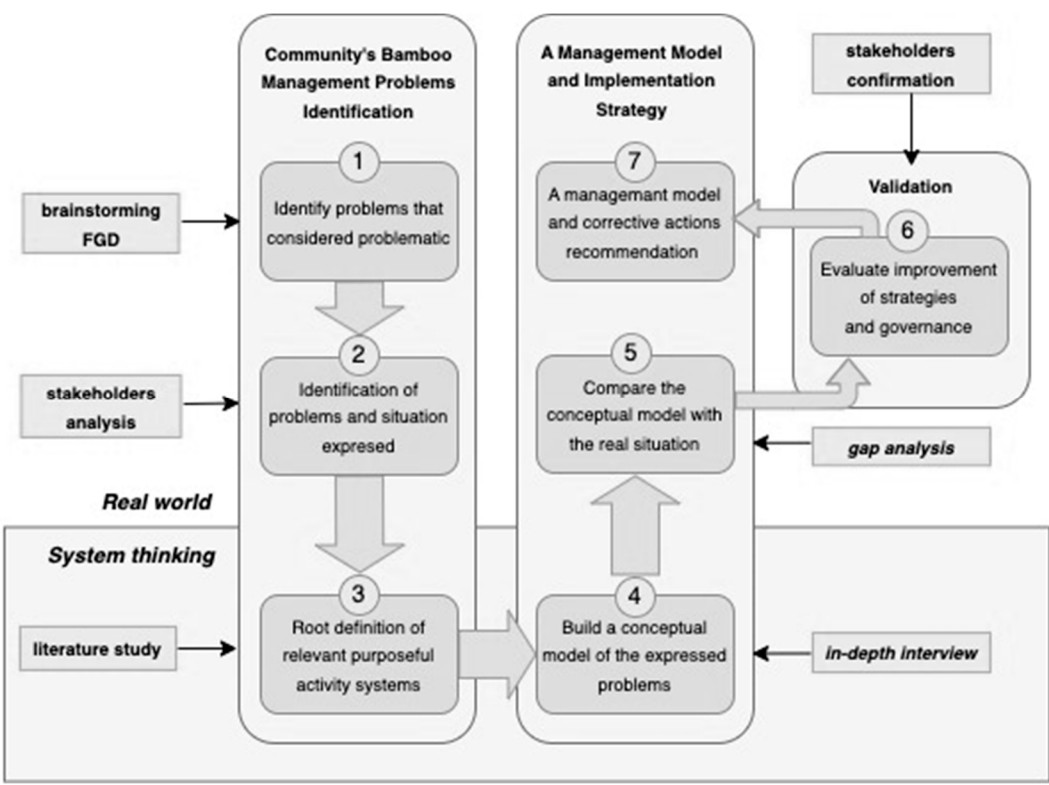

**Figure 2.** SSM framework used for study.

*2.3. Data Collection*

The data collection was conducted through a focus group discussion, brainstorming, a literature study, in-depth interviews, and direct field observations. This study and data collection took place from July 2019 to September 2022.

### 2.3.1. Focus Group Discussion (FGD) and Brainstorming

The first step of data collection was carried out through focus group discussion (FGD). Furthermore, a brainstorming session was conducted to identify, grab, and collect the problematic situation related to community bamboo resources management in Ngada Regency. The main goal is to obtain as much information as possible and capture problems in the management and utilization of community-owned bamboo. In this FGD, all data and information on the situation from various points of view and the actors involved in bamboo management and utilization were collected.

The FGD was held in the local government office of Ngada Regency. The discussion involved 40 participants that included heads of villages and sub-districts and representatives from the local government, central government, leaders of tribes, civil society, and private sectors.

Since the FGD included preeminent stakeholders related to community bamboo resources, an assessments of stakeholders' roles and their interests was also conducted. Furthermore, the stakeholder analysis approach was performed by identifying each stakeholder's interest, power, and support program. In comparison, other stakeholders who did not participate in the discussion identified their roles and responsibility on different occasions.

### 2.3.2. Literature Study

A literature study was carried out by collecting data and reports, regional planning documents, the central and provincial government's plans, statistical data, news, and publications related to the topic of study. It should be noted that very few academic journal articles exist on developing community-based bamboo as an industrial raw material in Indonesia. The documents and literature collected are used as references to support step three of the SSM methodology [18] to define the role of each group based on an analysis called CATWOE (customer, actor, transformation, world view, owner, and environmental constraints).

### 2.3.3. In-Depth Interview

The four steps of the framework to build and design a conceptual model that describes the activities from the root definition were formulated. A total of five respondents from different institutions of the expert group, consisting of two respondents from the local and central government, one respondent researcher, one from civil society, and one bamboo practitioner, were interviewed. The targeted group for an in-depth interview was the expert group consisting of policymakers, researchers, practitioners, and civil society who is in charge of and engaged in facilitating bamboo development, management, utilization, and research. All respondents in the expert group graduated with a bachelor's degree or higher.

### 2.3.4. Field Observations

Information and data were also collected from direct observations in the field during the authors' visits to the research area. This observational data collection was also conducted to examine potential community bamboo resources. Several field visits allowed the authors to interact directly with communities whose livelihoods are closely related to bamboo. The observation was also conducted on the associated stakeholders in their support of developing the supply of community-based bamboo as industrial-scale raw materials. In this case, the local government and the civil society or local non-governmental organization (NGO) assist the community in developing their bamboo resources and managing the bamboo's knowledges.

### *2.4. Data Analysis*

The data and information obtained in this study were processed and analyzed using the following approach.

### 2.4.1. Stakeholder Analysis

The stakeholder is the institutions, community groups, or individuals who are actively involved directly or indirectly in an activity or initiation; in this case, community-based bamboo development activities as raw materials for the bamboo industry. Stakeholders could have different roles and responsibilities inside or outside the organization, so it is necessary to identify the level of interest and how much power each stakeholder has, with the grouping shown in Figure 3 [20].

**Figure 3.** The quadrant of stakeholder analysis as adapted from Reed et al. [20].

1.  Individuals with high interest and high power are the most critical stakeholders and should be prioritized, informed regularly, and actively lobbied.
2.  Individuals with high interest and low power are the subject and beneficiaries to keep these stakeholders informed and check in with them regularly to ensure they are not experiencing problems.
3.  Individuals with high power and low interest keep stakeholders informed regularly and are potentially become players if shown more interest.
4.  Individuals with low power and low interest are the general public, who need to be advised, and who need a low level of reaction.

### 2.4.2. SSM Analysis

In the SSM framework, several analyses were also carried out by following seven stages and steps in the SSM framework, including a complex situation analysis poured into pictures, CATWOE analysis, conceptual model analysis, comparative analysis between conceptual and actual situation models, and systematic analysis of possible changes that will result in corrective actions and recommendations [19].

### 2.4.3. Gap Analysis

The approach using a gap analysis is carried out in the stages of the SSM framework used in the study. Specifically, gap analysis is performed for the conceptual model with the actual situation, which is the fifth stage of SSM. The gap analysis consists of four steps: (i) identifying the primary problem needs of the current situation, (ii) determining the ideal future or situation to be achieved, (iii) highlighting existing gaps that need to be filled, and (iv) modifying and implementing organizational plans to fill gaps [21].

## 3. Results

*3.1. Stakeholders and Their Roles in Bamboo Development*

The identification of stakeholders and actors related to the management of bamboo resources belonging to the community came from various elements: the community itself, i.e., both bamboo owners and bamboo artisans, institutions or economic groups, private parties, and the government. There are eighteen relevant stakeholders in community-based bamboo management and their roles and responsibilities, as presented in Table 1 below.

**Table 1.** Identified stakeholders, roles, and responsibilities.

| No. | Stakeholders | Role and Responsibility |
|---|---|---|
| 1. | Individual bamboo owners | Own private bamboo plants usually mixed with their mixed garden and produce poles and raw materials for the market and their own needs. |
| 2. | Communal bamboo owners | Own the communal bamboo plants inherited from their ancestors and tribe. |
| 3. | Bamboo crafters | Process bamboo into handicraft products, which is generally still on the scale of households and small industries. |
| 4. | Bamboo traders | Trade bamboo raw or as poles and bamboo products produced by craftsmen and the small-home industries level. |
| 5. | Cooperative, SMEs | Become an aggregator of bamboo products produced by craftsmen and small industries. |
| 6. | NGO/facilitator | Provide technical assistance or marketing access and are a facilitator in sustainable bamboo management for bamboo owners. |
| 7. | Bamboo strips industry | Provide support to farmers and craftsmen in the availability of processed bamboo raw materials. |
| 8. | Village Government | Provide support in data on bamboo potential in the region and regulatory support for bamboo management at the site level. |
| 9. | Ngada Planning and Development Office | Responsible for the regional planning development and the direction of supporting programs on community-based bamboo industry development. |
| 10. | Ngada Environment Office | Responsible for environmental management in Ngada Regency and its supporting programs and related policy support. |
| 11. | Ngada Industry Office | Responsible for the development of industry in Ngada Regency and its supporting programs and related policy support. |
| 12. | East Nusa Tenggara Com-Dev Office | Responsible for empowering rural communities, community institutions, and their supporting programs and related policy support. |
| 13. | East Nusa Tenggara Forestry and Environment Office | Responsible for managing the forestry and environmental sectors and their supporting programs and related policy support. |
| 14. | Ministry of Environment and Forestry | Responsible for forestry and environmental sector programs at the national level and policy and regulation support in the upstream sector. |
| 15. | Ministry of Village, Development of Disadvantaged Regions and Transmigration | Responsible for programs in the rural, disadvantaged regions and transmigration sectors at the national level and their support policies. |
| 16. | Ministry of Public Works and Settlement | Responsible for public works and housing sector programs at the national level and their support policies. |
| 17. | Ministry of Cooperative and SMEs | Responsible for the programs of the small, medium, and cooperative Usha sectors at the national level and its supporting policies. |
| 18. | Ministry of Industry | Responsible for industry and manufacturing development programs, from the small-scale industry sector to the high-technology industry and its supporting policies. |

The stakeholders' interests may be positively or negatively affected by the initiative. The stakeholders may have some needs and requirements that must be considered; there-

fore, this study should identify all the stakeholders at the beginning of the process. There are different ways to determine the stakeholder as the approval rights or decision-making. The integrated community-based bamboo management model needs to be approved by a government entity; therefore, the government becomes the primary stakeholder. Eighteen stakeholders related to the management and utilization of bamboo in Ngada Regency spread across subject and player groups. The Figure 4. shows that the stakeholders involved are directly involved in managing bamboo resources. The government, both central and local governments, are in high-power and high-interest positions. One non-governmental organization has a very high interest in bamboo, and this institution also has quite influential power in driving and initiating government programs.

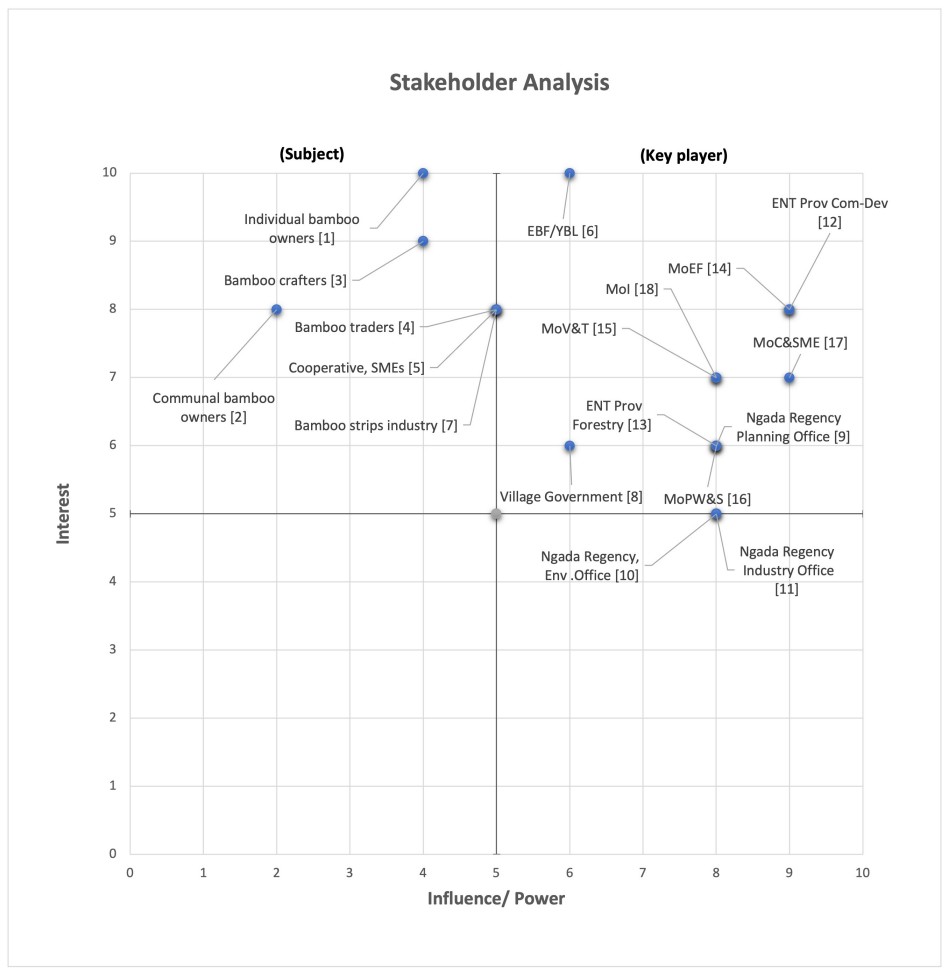

**Figure 4.** The results of stakeholder analysis.

### 3.2. Direct and Indirect Actors in Bamboo Development

The management and utilization of bamboo resources in Ngada Regency involve direct and indirect actors, as seen in Figure 5. The direct actors manage and utilize bamboo resources until they become products and are marketed to consumers. The direct actors include bamboo owners, both communally and individually, artisans, crafters, merchants, small- and medium-sized enterprises, the bamboo processing industry, civil society, and the Environmental Bamboo Foundation, which provides bamboo technical assistance.

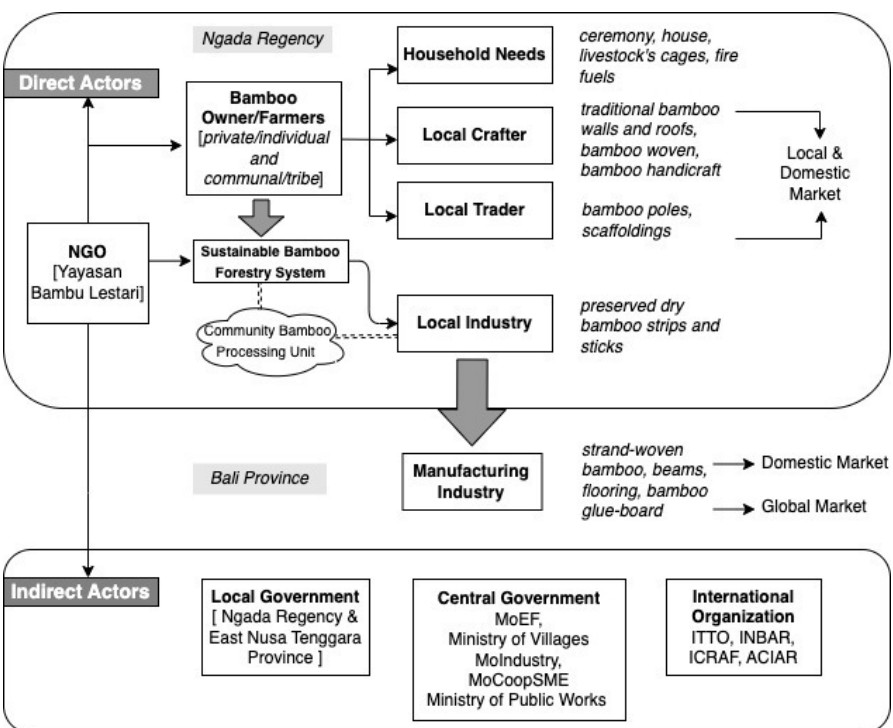

**Figure 5.** Direct and indirect actors in bamboo utilization in Ngada Regency.

Meanwhile, indirect actors are stakeholders who support the management of the community's bamboo resources in Ngada Regency, especially supporting activity programs, financial assistance, and infrastructure for its development. The indirect actors consist of local governments, central governments, and international institutions.

*3.3. Problematic Situation*

In general, the problematic situation in managing of potential community-based bamboo resources that has not been carried out optimally in Ngada Regency is that it has not been well connected between the parties and stakeholders in supporting the sustainability of management and utilization. The unconnected on-farm and off-farm activity, as well as unintegrated programs and regulations, help the management and utilization of community-based bamboo in Ngada Regency, as presented in Table 2.

**Table 2.** The identified problematic situation on community-based bamboo management in Ngada Regency.

| Sub-System | Identified Problematic Situations |
|---|---|
| on-farm activities | 1. data on potency and resources distribution are unavailable<br>2. lack of implementation of sustainable bamboo clumps management<br>3. lack of bamboo planting and propagation<br>4. lack of capacity building and institution on bamboo resources management<br>5. low accessibility to bamboo resources<br>6. land use change from bamboo to other purposes |
| off-farm activities | 7. household and small-scale bamboo utilization<br>8. lack of value-added products<br>9. lack of technology, innovation, and quality standards<br>10. lack of capacity building and institutions in bamboo processing and industry<br>11. lack of industrial culture<br>12. lack of investment in the bamboo industry |

**Table 2.** *Cont.*

| Sub-System | Identified Problematic Situations |
|---|---|
| market-consumer | 13. the uncertain market for bamboo products<br>14. unclear value-chain of bamboo products<br>15. lack of connection between on-farm and off-farm activities |
| technical facilitator | 16. lack of required number of field technical assistance<br>17. lack of training and capacity building for field facilitators<br>18. low interest from the young generation to become field facilitators |
| program support | 19. unconnected on-farm and off-farm of government programs<br>20. unintegrated program between local and central government<br>21. unsustained and partial programs implementation |
| regulation support | 22. lack of bamboo management regulation from the village to the national level<br>23. unavailable transparent governance of bamboo management and utilization |

*3.4. Expressing the Problem Situation*

At this stage, the problem situation is expressed in rich pictures based on the study of the unstructured problematic situation encountered. Rich illustrations are built from the problems observed and encountered related to the research topic. The management and utilization of bamboo resources owned by the community in Ngada Regency show a complex situation drawn into rich pictures, as shown in Figure 6. The rich images depict the activity process of each institution and the parties involved. In the picture, it can be seen that the stakeholders have carried out activities that have been ongoing, which are shown by solid lines and their relationship with other stakeholders. The dashed lines depict activities that are yet to be implemented.

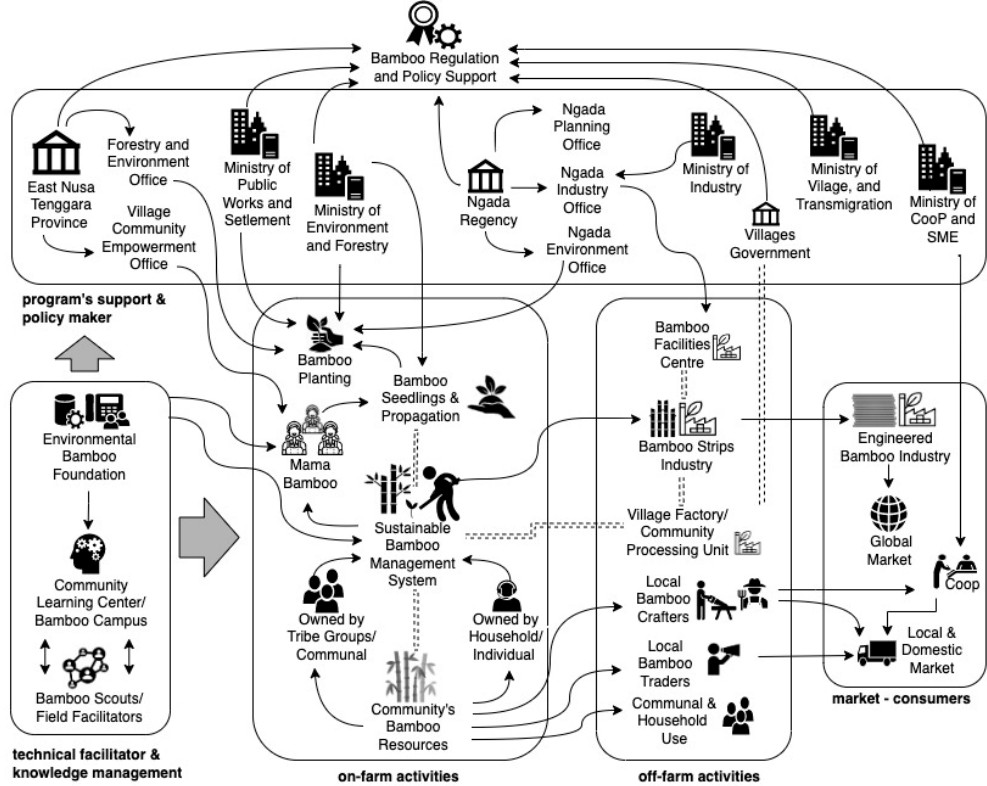

**Figure 6.** Rich picture situation of community-based bamboo management in Ngada Regency, East Nusa Tenggara.

From the picture, it can be seen that the stakeholders involved are divided into several activity clusters, namely on-farm activities, off-farm activities, market and consumers, technical facilitators and knowledge managers, regulators/policymakers, and program support. A non-governmental organization in Ngada plays an essential role as a bridge that connects the governments as regulators and programs help with on-farm and off-farm activities.

### 3.5. Root Definition Formulation

This stage was carried out by building a problem definition related to the problematic situation that occurs or was referred to as root definition formulation. Formulated using the CATWOE analysis technique critically, or containing CATWOE elements or more concisely expressed in the PQR formulas: what to do (P), how to do it (Q), and why do it (R) [18]. The compilation of root definitions plays an essential role as a basis for conceptual model creation, as described in Table 3.

**Table 3.** The results of the CATWOE analysis.

| CATWOE Elements | Description |
|---|---|
| Customer | The parties who benefit or potentially become victims of the management model, or the beneficiaries, may also be the aggrieved parties of the model developed; in this case, community and bamboo business actors (bamboo owners, craftsmen, and industry). |
| Actors | In this case, the parties that perform essential activities or who will carry out important activities in realizing sustainable community bamboo management are the stakeholders involved: local governments, central governments, non-governmental organizations, and downstream bamboo processing industries. |
| Transformation | The process built in the integrated management model through community-based integrated programs and active participation. Includes building the active participation of stakeholders directly Involved and synchronized programs in the sustainable management of community bamboo. |
| Weltanschauung/World view | The ideal goal to be achieved is to realize the strategy model for managing and utilizing bamboo that provides sustainable benefits, economically and socially. |
| Owners | The parties who can or have the power to stop or change the process, in this case, the local government, the central government, the manufacturing bamboo industry, and NGOs. |
| Environmental Constraints | The obstacles that may arise from the scope of the model built in this case, such as taking into account the potential of bamboo resources, and market and business environment uncertainty. Moreover, the unsupported regulations, uncertain data on the potency of bamboo resources, uncertain market of the bamboo product, and land use changes (bamboo conversion). |

Based on the conditions mentioned in Table 3, the root definition of the existing situation is as follows: a strategy model for managing and utilizing bamboo that provides economically and socially sustainable benefits (W) for bamboo business actors (bamboo owners, craftsmen, and industry) (C), while contributing to a sustainable environment through community-based integrated programs and active participation (T), of the stakeholders involved (A) by taking into account the potential of bamboo resources and market and business environment uncertainty (E). Strategies and programs are built by the central and local governments and implemented by the relevant regional apparatus and local NGO (O).

### 3.6. Compared a Conceptual Model with the Actual Situation and Action Plan

This section is a series of stages 5, 6, and 7 in the SSM framework used in this study. In stage 5, the conceptual model was compared with the actual situation using gap analysis. The analysis was carried out to see the gap between the ideal conceptual model and the current actual conditions and to draw up an adaptive and appropriate agreed-upon action plan and implemented by the stakeholders involved (Table 4).

**Table 4.** The results of the CATWOE analysis.

| Sub-System | Conceptual Model | Real Situation and Condition | Gap | Corrective Action Plan |
|---|---|---|---|---|
| on-farm activities | <ul><li>data on potency and resources distribution available</li><li>implementing sustainable bamboo clump management</li><li>increasing bamboo planting and propagation</li><li>increasing the capacity building and institution of bamboo resources management</li><li>improving accessibility to bamboo resources</li></ul> | <ul><li>data on potency and resources distribution are unavailable</li><li>lack of implementation on sustainable bamboo clump management</li><li>lack of bamboo planting and propagation</li><li>lack of capacity building and institution on bamboo resources management</li><li>low accessibility to bamboo resources</li><li>land use change from bamboo to other purposes</li></ul> | <ul><li>need for survey and inventory on bamboo resources and distribution required</li><li>need for the implementation of sustainable bamboo clump management</li><li>no direction on planting program and support on bamboo propagation</li><li>no training program and strengthening institution for the community and stakeholders involved</li><li>need for infrastructures and access</li><li>bamboo still undervalued</li></ul> | <ul><li>conducting surveys and inventory on bamboo resources and distribution starting at the village level</li><li>socializing and implementing the sustainable bamboo management system</li><li>conducting bamboo propagation and planting bamboo in degraded land and watershed area</li><li>conducting training on the sustainable bamboo management system, bamboo field school.</li><li>developing infrastructure to access bamboo resources</li><li>socializing bamboo utilization and its ecological roles</li></ul> |
| off-farm activities | <ul><li>improving household and small-scale bamboo utilization</li><li>increasing the value-added bamboo products</li><li>improving the technology, innovation, and quality standards</li><li>increasing the capacity building and institution of bamboo processing and industry</li><li>developing the industrial culture</li><li>expanding the investment in the bamboo industry</li></ul> | <ul><li>household and small-scale bamboo utilization</li><li>lack of value-added products</li><li>lack of technology, innovation, and quality standards</li><li>lack of capacity building and institutions in bamboo processing and industry</li><li>lack of industrial culture</li><li>lack of investment in the bamboo industry</li></ul> | <ul><li>no intensive assistance for small-scale bamboo utilization</li><li>no assistance and training on diversification products</li><li>need for support on suitable technology and innovation</li><li>need for training program and technical assistance</li><li>requiring a transformation from subsistence to generating income</li><li>lack of investment in the bamboo industry</li></ul> | <ul><li>assistance program for small-scale bamboo utilization</li><li>support program on suitable technology and innovation of bamboo processing at community level</li><li>training program on bamboo processing and institution strengthening</li><li>increasing bamboo utilization for income generation</li><li>facilitating investment and ease of doing business</li></ul> |

**Table 4.** *Cont.*

| Sub-System | Conceptual Model | Real Situation and Condition | Gap | Corrective Action Plan |
|---|---|---|---|---|
| market-consumer | • developing market for bamboo products<br>• strengthening the value chain of bamboo products<br>• connecting on-farm and off-farm activities | • the uncertain demand for bamboo products<br>• unclear value-chain of bamboo products<br>• unconnected on-farm and off-farm activities | • need for support on bamboo business development<br>• need for establishing a solid and sustainable value chain<br>• need for promotion and campaign on bamboo products | • developing bamboo business and cooperation with off-taker<br>• connecting and strengthening the bamboo products value chain from on-farm to off-farm<br>• promoting and campaigning bamboo products |
| technical facilitator | • increasing the number of field technical assistance<br>• increasing the number of the training program and capacity building for field facilitators<br>• raising awareness and interest of the young generation to become field facilitators | • lack of the number of field technical assistance<br>• lack of training and capacity building for field facilitators<br>• low interest from the young generation to become field facilitators | • need for public awareness of the importance of bamboo utilization<br>• no curriculum and module for bamboo development | • conducting several trainings of the trainee as technical assistant in the field<br>• developing bamboo curriculum and module for formal and informal capacity building<br>• increasing public awareness of bamboo utilization and its roles |
| program support | • connecting the government program for on-farm and off-farm<br>• integrating programs between local and central government | • unconnected on-farm and off-farm of government programs<br>• unintegrated program between local and central government<br>• unsustained and partial programs implementation | • no integrated regional strategy<br>• need for multi-year program support<br>• need for a synchronized program between the local and central government | • developing an integrated regional plan for bamboo development<br>• developing a multi-year program on bamboo development<br>• synchronizing the programs of central and local government and the agency within |
| regulation support | • developing the bamboo management regulation from the village to the national level.<br>• improving the governance of bamboo management and utilization | • lack of bamboo management regulation from the village to the national level<br>• unavailable transparent government of bamboo management and utilization | • no assistance to develop village regulations to support bamboo<br>• need for integrating bamboo development into the regional management plan | • socializing and assisting the establishment of villages regulation on bamboo<br>• incorporating bamboo development in the regency management plan<br>• developing rules and policy at the regency level |

## 4. Discussion

Bamboo has great potential, with a wide range of products that can be produced and providing environmental services, making bamboo a potential type recommended in the integrated development of rural areas [22]. In Ngada Regency, bamboo is primarily used for subsistence and daily life purposes, and the bamboo products produced by the community are mainly weaved bamboo and handicraft products. Bamboo had not been a commodity for commercial or industrial use. Existing bamboo resources must be adequately managed, and the community's perception of bamboo remains low because they have yet to see bamboo provide significant economic value. Furthermore, utilization is still limited to simple processing with little added value [23]. This situation was exacerbated by the conversion of the function of community bamboo gardens into other purposes.

The assumptions from the SSM framework used in this study to build an integrated management model in the management and utilization of community-based bamboo in Ngada Regency are as follows: (i) there are problems in the management and utilization of communities' bamboo in Ngada Regency, (ii) the interpretation of the problem by the parties varies according to their respective points of view, (iii) human factors and the activities they carry out play an important role, (iv) problem solving is carried out with a creative and intuitive approach, and (v) the result is learning about and gaining a better understanding of the problems faced [24].

In natural resource management, identifying stakeholders is a crucial stage. Billgreen and Holmes [25] identified the aims of stakeholder analysis in natural resource management as follows: (i) identify and categorize the stakeholders that may have influence, (ii) develop an understanding of why changes occur, (iii) establish who can make changes happen, and (iv) determine how to best manage natural resources. The study results produce appropriate corrective actions and the strategic recommendation formulated from conceptual models and existing actual conditions on integrated sustainable bamboo management in Ngada Regency, which was drawn up and agreed upon by involved stakeholders.

In 2012, an industry in Ngada Regency processed bamboo into raw materials for laminated bamboo products, which requires 3 tons of bamboo per day [6]. Alternative uses of bamboo based on industry need more raw materials continuously; therefore, it requires a management and harvesting system with sustainability principles [26]. Applying a management and harvesting system with sustainable principles is a must in using bamboo resources for large-scale industrial needs and general use. The main challenge in the on-farm section to ensure the sustainability of clump maintenance is also to increase the productivity of bamboo clumps [27]. Preserved bamboo groves will also provide ecological functions and more optimal environmental services [28]. Community-based bamboo management and utilization is an opportunity, especially in developing rural areas towards sustainable development [29]. Studies related to community-based bamboo management in Indonesia have been carried out, including management on the banks of rivers and catchment areas (upstream areas) [30]. Furthermore, community-based bamboo management, in general, uses a hybrid plant system (agroforestry) [31], which is usually combined with local knowledge gained for generations with local wisdom [32]. According to the economic analysis, once bamboo clumps mature, culms can be harvested all year. Monocrop bamboo cultivation may be appropriate for restoring degraded lands and benefiting large-scale charcoal producers or where farmers have sufficient land to allow its establishment. On the other hand, small-scale farmers could benefit from bamboo intercropping systems for at least three years by increasing system productivity, diversifying income streams, and ensuring environmental sustainability [33].

The biggest challenge of on-farm management is that bamboo is still treated as a wild plant rather than as a cultivation commodity. Therefore, bamboo planting is yet to be commonly carried out by the community. Even though bamboo is one of the potential types that can be used in eco-restoration activities from degraded land [34], land restoration can also contribute to the community's economy [35]. Another environmental function of bamboo planting is the use of bamboo for erosion control and slope stabilization [36], as

well as bamboo vegetation bringing the stability of riverbank area [37]. Related to climate change, bamboo planting and restoration will contribute to emission reduction due to its carbon sequestration [38]. The community-based bamboo planting and reforestation offer opportunities for biomass production and carbon farming [39].

From the description of functions and benefits of bamboo planting, the challenges of cultivating bamboo can be answered, where in addition to getting economic benefits, the community experiences improved environmental ecology. Furthermore, the three main points of the on-farm issue need to be considered, i.e., sustainable bamboo clump management, sustainable harvesting, and bamboo planting, as sequences activities to ensure sustainable bamboo resources management. In Ngada Regency, the sustainable bamboo management mechanism is only applied to the supply of preserved bamboo strips industry. Meanwhile, using bamboo for other products has not yet implemented a sustainable bamboo harvesting system. Therefore, the technical assistance and knowledge transferred related to sustainable bamboo management play a significant role. The role of non-governmental organizations or civil society in providing assistance and management of technical knowledge about bamboo and its processing for the community and other related stakeholders becomes a driver in building the enabling conditions.

From the identification of problematic situations and challenges, it is not only from on-farm and off-farm sectors, but also the technical assistance activities needed to support the community-based bamboo development in Ngada Regency. Therefore, the support of programs, regulations, and policies from the village level to district, sub-national, and national levels is also required. Furthermore, a clear and integrated management system will support the good governance of sustainable utilization of community-based bamboo, as has been achieved in Southern China and Anji County, where rural development has been acquired from the contribution of bamboo forestry governance and management [40,41].

In implementing community-based bamboo management, another crucial supporting factor is gender equality. In this case, women's roles in managing and utilizing bamboo cannot be separated. In Ngada Regency, women play an essential in contributing to household activities, including activities in their bamboo garden [42]. At the research site, women play a role in the manufacturing and providing of bamboo seedlings, with the assistance of local non-governmental organizations and local government funding support [43]. These women's activities contribute to and drive the on-farm sector. While in off-farm sectors, Ekawati et al. [24], in the previous research, discovered that individual bamboo crafters, primarily women on a household scale, largely are responsible of producing bamboo woven products in Ngada Regency. The same thing was also found in other areas, where the bamboo handicraft sector, specially woven products, are produced by women, such as in Tasikmalaya, West Java [44], Bangli, Bali [36], and Lake Toba, North Sumatera [45].

The feasibility study of potential bamboo businesses in Ngada Regency produced recommendations indicating that available bamboo has a high potential for use in modern products, such as the laminated and construction bamboo industries [46]. According to the study, bamboo has been used in commercial products in the downstream sectors, but primarily for local and still limited markets. Proper planning and analysis are fundamental in using bamboo in the off-farm sector, especially in community-based development. Approach using participatory rural appraisal [47] and SWOT analysis [48] become a tool and approach that can be used in building a community-based SME bamboo business strategy [49]. The business development strategy will run with the establishment of a solid and sustainable value chain and market [3,50]. Other challenges from the off-farm sector in using bamboo were the need for product development and innovation skills. Therefore, training and capacity building in the processing and development bamboo products is crucial. Another supporting factor that ensures the model management of community-based bamboo resources management is run and sustainable is regulations support. The recommendations resulting from this study are also part of regulatory and policy support in community-based bamboo resource management. A management model based on the actual conditions and problematic situations faced by stakeholders and actors

will certainly provide suitable recommendations. The resulting integrated management model is a process of discussion, confirmation, and agreement among the stakeholders.

## 5. Conclusions

The potential bamboo resources owned by the Ngada community are yet to be managed and utilized appropriately and optimally. The roles and responsibilities of the stakeholders involved still need to be clarified. The main issue of this situation is the need for an integrated program that connects between on-farm and off-farm sectors, as well as the lack of clear roles and responsibilities among the stakeholders involved.

Based on the current conditions, the system thinking approach was used in developing a conceptual strategy model that will be compared with actual conditions with gap analysis. The formulation of root definition based on the existing situation is a strategy for managing and utilizing bamboo that provides sustainable benefits (W) for business actors (bamboo owners, craftsmen, and industry) economically and socially (C) while contributing to a sustainable environment through integrated programs (T) and community-based and active participation of stakeholders (A) by taking into account the potential of bamboo resources and market and business environment uncertainty (E). Strategies and programs are built by the central and local governments and implemented by the relevant regional apparatus and NGO (O).

The formulation of the management strategy as a corrective action plan, which is the agreement among parties involved as follows: (i) on farm activity: survey and inventory of bamboo resources, implementation of sustainable bamboo management systems, bamboo planting and cultivation, and upstream infrastructure support, (ii) off-farm activity: bamboo processing with added value, support of appropriate technology and innovation, capacity and skill building, and strengthening of people's economic institutions, (iii) markets and consumers: development of bamboo business cooperation, strengthening the value chain, and promotion and campaigning of the use of bamboo products, (iv) technical assistance: bamboo field schools and increasing public awareness about the role and fusion of bamboo, and (iv) program and regulatory support: synergy of government programs from the village, district, provincial, and national level regulations in community-based bamboo development.

**Author Contributions:** Conceptualization, L.K., R.S., M. and D.E.; data curation, D.E.; formal analysis, D.E., R.S. and M.; funding acquisition, L.K.; investigation, D.E.; methodology, D.E., R.S. and M.; project administration, L.K. and D.E.; resources, L.K. and D.E.; Software, D.E. and M.; Supervision, L.K., R.S. and M.; validation, L.K., R.S. and M.; visualization, D.E., R.S. and M.; Writing—original draft, D.E., L.K., R.S. and M. All authors have read and agreed to the published version of the manuscript.

**Funding:** The publication was funded by a Doctoral Research Grant from the Ministry of Education and Higher Education, Indonesia: No.0267/E5/AK.04/2022 and No. 001/E5/PG.02.00PT/2022. The study and data collection were supported by project FST/2021/161 and FST/2016/141; collaboration research between the Ministry of Environment and Forestry (MoEF) of Indonesia the Australian Center for International Agricultural Research (ACIAR).

**Data Availability Statement:** The authors confirm that the data supporting the findings of this study are available within the article. The raw data supporting this study's results are available from the corresponding author upon reasonable request.

**Acknowledgments:** This manuscript has been derived from doctoral research and collaborative research actions conducted in Ngada Regency, East Nusa Tenggara. Thank you for the support and collaboration; to the Environmental Bamboo Foundation (EBF) of Bali and Flores, the local government of Ngada Regency; the community groups in Ngada Regency; bamboo farmers, mama's bamboo, bamboo crafters, and all of our researchers team in the projects. Finally, we thank the three anonymous reviewers or their insightful comments on the earlier versions of the manuscript and MDPI journal editors who help on the process of publishing this paper.

**Conflicts of Interest:** The authors reported no potential conflict of interest.

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
