# Peer review of "A Model of Integrated Community-Based Bamboo Management for the Bamboo Industry in Ngada Regency, East Nusa Tenggara, Indonesia"

_sustainability, doi:10.3390/su15020977_

Round 1
Reviewer 1 Report
This is a well-written article on a topic of relevance to all. There is a need to add a few more references on the application and potential of bamboo. The article can be further strengthened by correcting a few grammatical errors, especially the conclusion paragraph should be rewritten to avoid grammatical mistakes. otherwise, the article can be accepted for publication with minor revisions.
Author Response
Dear Sir/Madam
Thank you very much for the review of our manuscript. As per your input, we revised the manuscript. Your fruitful review and comments are appreciated.
Best regards,
Desy

Reviewer 2 Report
The study approach was appropriate for the research objectives. The authors provided an array of conceptual models to depict stakeholders' roles, responsibilities, interests, power, and their connections in pursuit of a sustainable bamboo management and industry. The two major downsides of the paper is its linguistic flow and the conclusion section. The The authors need to perform wordsmithing to improve clarity of their outstanding research. Additionally, the conclusion was a rehash of the results with no clear policy implications.
Please see below for specific comments:
Line 22 : replace potency with potentials
Line 26 : replace ; with :
Line 28 : manufacture bamboo industry is meaningless. please revise
Line 43: replace “encourages” with “underpin”
Line 75: strengthen what?
Line 95: this sentence is incomplete “as well as car and”
Line 131: correct in-an
Line 157-159: are the heads of villages and leaders of villages different?
Line 177: I think it should be “definition was formulated”
Line 183-184: did you mean bachelor’s degree? Is there any education institution higher than university?
Line 187: authors’
Line 198: revise “as a following;”
Line 200: specify as persons or institutions instead of “everyone”
Line 205-206: revise to “should be prioritized, informed regularly, and actively lobbied”
Line 227: replace implementing with implement
Line 230-233: should move to discussions
Line 236: put : in front of elements
Table 1:
-Check spelling of “and” between Role and Responsibility
-Bamboo traders – correct home house
- Revise the Role and Responsibility of Ngada Planning and De-velopment Office
- Ministry of Environment and Forestry: policy support instead of just support
- # 14: the name is Ministry of Villages, Development of Disadvantaged Regions, and Transmigration
- # 17: check “and” after the Cooperative
Line 243: revise the sentence “needs one’s requirements”
Line 245: “identify evaluated the” is not meaningful
Line 250: delete the “governments” at the end of this sentence because it’s already mentioned
Line 262-265 is the same as 269-271. Please delete one
Line 275-279: I think this a repeat of the methods. I suggest you begin section 3.3 with Line 180- In general … and highlight lack of coordination among stakeholders as the major bottleneck.
Table 2:
- #2: implementation of ….
- # 15: revise to “lack of connection between ….”
- # 16: maybe “lack required number of ….
Line 297-298: revise the sentence into a meaningful one
Line 324: use ‘compared’
Table 4:
- Use ‘household’ throughout.
- Correct implementation on to “implementation of”
- off-farm activities GAP: revise last option to “lack of investment in bamboo industry”
- off-farm activities CAP: replace last one with “facilitate investment and ease-of-doing business”
- market-consumer GAP: use ‘establish’ in point #2
- market-consumer- Real Situation and Condition: remove between in the point #3
- program support- Real Situation and Condition: use ‘unsustained’
Line 344: remove ‘messy’
Line 372: remove ‘and’
Line 373: remove ‘service’
Line 279: sentence not clear
Line 389-393: I do not get the intended message
Line 397-400: Sentence not clear
Conclusion: The conclusion should provide the practical and policy implications of the findings instead of repeating the results.
Author Response

(The authors gave the same response as above.)

Reviewer 3 Report
In general, this paper is quite good in terms of content, but there are several issues related to grammar that must be corrected. As a reviewer, I immediately corrected this article with Grammarly's help. Initially there were around 390 issues that needed to be fixed with a score of 78. But after I fixed them, the score had reached 99. So in terms of language, they were sufficient to be published. Furthermore, I have also analyzed the similarity (plagiarism checker) of about 13%. So this article deserves to be published. However, authors must correct or reformat according to the journal template so that it fits the specified requirements. I immediately corrected the errors with the consideration that the content of the article was very good and helped the authors to fix it immediately based on the draft that we attached to my review report. Thank you

Author Response

(The authors gave the same response as above.)
